# Enhanced Electrochemical Water Oxidation Activity by Structural Engineered Prussian Blue Analogue/rGO Heterostructure

**DOI:** 10.3390/molecules27175472

**Published:** 2022-08-25

**Authors:** Xiuyun An, Weili Zhu, Chunjuan Tang, Lina Liu, Tianwei Chen, Xiaohu Wang, Jianguo Zhao, Guanhua Zhang

**Affiliations:** 1Department of Mathematics and Physics, Luoyang Institute of Science and Technology, Luoyang 471023, China; 2State Key Laboratory of Advanced Design and Manufacturing for Vehicle Body, College of Mechanical and Vehicle Engineering, Hunan University, Changsha 410082, China; 3College of Physical & Electronic Information, Luoyang Normal University, Luoyang 471934, China

**Keywords:** oxygen evolution reaction, Prussian blue analogue, graphene, heterostructure

## Abstract

Prussian blue analogue (PBA), with a three-dimensional open skeleton and abundant unsaturated surface coordination atoms, attracts extensive research interest in electrochemical energy-related fields due to facile preparation, low cost, and adjustable components. However, it remains a challenge to directly employ PBA as an electrocatalyst for water splitting owing to their poor charge transport ability and electrochemical stability. Herein, the PBA/rGO heterostructure is constructed based on structural engineering. Graphene not only improves the charge transfer efficiency of the compound material but also provides confined growth sites for PBA. Furthermore, the charge transfer interaction between the heterostructure interfaces facilitates the electrocatalytic oxygen evolution reaction of the composite, which is confirmed by the results of the electrochemical measurements. The overpotential of the PBA/rGO material is only 331.5 mV at a current density of 30 mA cm^−2^ in 1.0 M KOH electrolyte with a small Tafel slope of 57.9 mV dec^−1^, and the compound material exhibits high durability lasting for 40 h.

## 1. Introduction

The energy problems caused by traditional fossil energy have become a major issue of global concern, and countries around the world have devoted many efforts for developing and storing renewable energy resources to alleviate this problem [1,2,3]. China formally proposed the “double carbon” objective in 2020, with the goal of reaching a peak in carbon output by 2030 and carbon neutrality by 2060. Therefore, it is imperative to develop and utilize sustainable energy to minimize carbon emissions [4,5]. Hydrogen has a high energy density and generates zero carbon emissions, making it perfect for renewable energy production [6,7]. Among various approaches, hydrogen generation using water electrolysis is an efficient way to obtain pure hydrogen energy [8,9]. Compared with the hydrogen evolution reaction, the oxygen evolution reaction involves the four electrons transmission, which is a more complicated process and energy-intensive process, limiting the efficiency of electrolysis-based hydrogen production [10]. Thus, it is necessary to develop and design efficient and stable oxygen evolution catalysts, which is one of the global interests for current research regarding hydrogen production.

The Fe-based transition-metal catalysts present low-cost relative to current commercial noble metal catalysts and have high catalytic activity and stability under alkaline conditions [11,12]. It is worth noting that Prussian blue-like material is a typical metal-organic framework structure material with an open skeleton, adjustable components, and a large number of unsaturated coordination atoms [13,14]. Its derivatives, such as sulfides and phosphides often exhibit superior catalytic activity for oxygen evolution [15,16]. However, the preparation of the derivatives frequently requires high-temperature treatment, which causes large energy consumption. PBAs have also been reported to be used directly as electrocatalysts. For example, D. Jason Riley et al. used anodic oxidation to create NiCo@A-NiCo-PBA core-shell structure [17], while Zhu et al. increased OH^−^ adsorption at the reaction site by modulating the morphology and components of FeCoNi ternary PBA composites [18]. However, PBA has a poor charge transfer capacity, which restricts its intrinsic catalytic efficacy. Loading PBA materials on highly conductive materials is an effective way to improve their charge transfer capability [19,20].

Graphene-based materials with excellent physicochemical properties have been intensively utilized in environmental [21], and catalytic applications [22]. It is worth noting that surface defects and edge atoms of graphene with electrochemical activity lead to the adsorption of tiny molecules (e.g., CO_2_) or redox reactions [23,24]. However, graphene exhibits poor intrinsic activity in the four-electron transferred OER process. The theoretical and experimental literature research reveals that graphene surface modification or doping improves local charge transfer. In particular, the local charge transport changes the surface properties of the modified material which facilitates electrochemical reactions [25,26,27]. Given this, it is envisaged that combining graphene with high-activity PBA composites will result in high-activity OER catalysts. Specifically, the fabrication of graphene-based heterostructures provides the following advantages. The surface of graphene oxide (GO) is rich in negatively charged functional groups, which can serve as anchor sites for the domain-limited growth of catalyst materials, while the charge transfer of the materials and the charge state density of the reaction sites can be modulated by the interfacial charge transfer interaction between heterostructures [28].

By constructing PBA-based heterostructures, this work investigates the effects of interfacial interactions on material properties. Graphene and NiFe bimetallic PBA composites were prepared using a facile co-precipitation method. The introduction of graphene provided anchoring-domain-limited growth sites for the PBA nanoparticles. According to the electrochemical test results, adding graphene increased the PBA material’s electrochemically active area and the number of exposed reaction sites. Interfacial charge transport between PBA and reduced graphene oxide (rGO) carriers was discovered by materials’ physical and electrochemical characterization, which not only enhanced the charge transport capacity of the composite but also modulated the activity of the reaction sites. The overpotential of the material was only 331.5 mV at a current density of 30 mA cm^−2^ with the Tafel slope of 57.9 mV dec^−1^ in 1.0 M KOH electrolyte, and the composite performed continued oxygen evolution for 40 h at a current density of 30 mA cm^−2^ under alkaline conditions.

## 2. Experimental Section

### 2.1. Materials

Graphite (C, SR), sodium nitrate (NaNO_3_, AR, ≥99.0%), potassium permanganate (KMnO_4_, AR, ≥99.5%), potassium ferricyanide (K_3_Fe(CN)_6_, AR, ≥99.5%), nickel chloride hexahydrate (NiCl_2_⋅6H_2_O, AR, ≥98.0%), and iron trichloride hexahydrate (FeCl_3_⋅6H_2_O, AR, ≥99.0%) were purchased from Sinopharm Chemical Reagent Co., Ltd. Hydrogen peroxide aqueous solution (H_2_O_2_, CR, 35%), sulfuric acid (H_2_SO_4_, CR, 95.0~98.0%), hydrochloric acid (HCl, CR, 36.0~38.0%) were purchased from Tianjin Chemical Reagent Co., Ltd. Nafion solution (5 wt%) was obtained from DuPont company. All of the compounds were used as they were acquired, with no additional purification. The water used in the experiment was distilled (18.25 MΩ cm). The hydrophilic carbon cloth (CC, WS1101) with a thickness of 0.36 mm was purchased from Taiwan CeTech Co., Ltd.

### 2.2. Material Synthesis

Oxidized graphene (GO) was prepared by acid-oxidation of natural graphite powder through modified Hummer’s method, which has been reported previously [29]. The PBA/rGO composite was prepared with the co-precipitation method. First, 1 mL of 0.5 mM K_3_Fe(CN)_6_ solution was slowly poured into 10 mL of 0.8 mg/mL GO solution, sonicated for 10 min, and centrifuged at 6000 rpm. Second, the supernatant was removed, and the resulting precipitate was dispersed in water. Third, 2 mL of 0.5 mM NiCl_2_ solution was quickly added to the above dispersion, stirred rapidly for 5 min, and left to stand for 25 min resulting brown mixture. Then, centrifugation was used to obtain a composite of PBA and GO (PBA/GO), which was then washed three times with deionized water. To improve the conductivity of GO composite, 1 mL of 88 mg mL^−1^ of the ascorbic acid solution was added after PBA/GO composite shake-dispersed in 20 mL of deionized water, then left to react for two hours in a water bath at 90 °C. After washing several times with deionized water, the PBA NiFe 2-1/rGO composite nanomaterial was obtained by freeze-dried precipitate overnight. The preparation of the PBA material was through a similar process except that the GO solution was not added, and the GO reduction step was not done in a 90 °C water bath. The molar ratios of Ni and Fe atoms in the K_3_Fe(CN)_6_ and NiCl_2_ precursors were adjusted to obtain Ni/Fe = 1:1, 2:1, 3:1, 4:1 materials labeled as PBA NiFe 1-1, PBA NiFe 2-1, PBA NiFe 3-1, and PBA NiFe 4-1, respectively. PBA NiFe 0-1 material was obtained by replacing the NiCl_2_ precursor species with FeSO_4_.

### 2.3. Preparation of Electrodes

Electrochemical electrodes were formed by dropping the samples onto the CC. The hydrophilic CC (1 × 1.2 cm^2^) was typically washed with acetone and ethanol before drying in air at ambient temperature. The obtained powder material (10 mg) was then ultrasonically disseminated in 0.5 mL of ethanol and water (1:1) combined with 30 μL of Nafion solution for one hour to prepare a homogenous catalyst ink solution. Then, 25 μL of ink was drop-coated onto the dry CC substrates in an area of 1 cm^2^ with a loading of 0.5 mg cm^−2^ and dried overnight in the air at room temperature.

### 2.4. Material Characterization

Scanning electron microscopy (SEM, Sigma HD, Carl Zeiss AG, Oberkochen, Germany) with a 10 kV accelerating voltage was used to examine sample morphologies. X-ray diffraction (XRD, Rigaku Ultima IV, Rigaku, Japan) with a Cu Kα radiation source (λ = 0.15418 nm) was carried out to determine the phase structure and crystallinity of the samples. The XRD patterns were obtained with a scan rate of 5° min^−1^ in the 10 to 75° range. The chemical structures of samples were obtained using Raman spectra (WITec Alpha-300R, WITech GmbH, Ulm, Germany) with the wavelength of 532 nm. X-ray photoelectron spectroscopy (XPS, PHI-Vesoprobe 5000 III, ThermoFischer, America) equipped with a monochromatic Al Kα ray source was taken to analyze the chemical band information of the samples, and binding energies were standardized with the C 1s peak at 284.8 eV.

### 2.5. Electrochemical Measurements

On the electrochemical workstation, the electrocatalytic performance of PBA NiFe x-1/CC electrodes was tested using a typical three-electrode setup (CHI660E). The working electrode (WE) was a PBA NiFe x-1/CC, the counter electrode (CE) was a graphite rod and the reference electrode was an Ag/AgCl (3 M KCl). The standard electrode potential of the freshly calibrated reference electrode was 0.2 V; the subsequent conversions of the RHE electrode potential were relative to this value. According to the equation: $E_{\mathrm{RHE}}=E_{\mathrm{Ag}/\mathrm{AgCl}}+0.2+0.059\times pH$, all measured potentials were transformed to the reversible hydrogen electrode (RHE). The pH of the electrolytes was estimated to be 14 for 1.0 M KOH.

The PBA NiFe x-1/CC electrodes were treated to 20 cycles of continuous cyclic voltammetry (CV) activity before the OER measurements until reproducible voltammograms were obtained. The OER activity was tested through linear sweep voltammetry (LSV) in 1.0 M KOH at a scan rate of 5 mV s^−1^ in the potential range of 1.126 to 1.826 V vs. RHE. The electrochemical impedance spectroscopy (EIS) analysis was carried out at open circuit voltage with 5 mV ac perturbation in the frequency range from 1 kHz to 0.01 Hz. The electrocatalytic stability was measured by chronopotentiometry (CP) over the current density of 30 mA cm^−2^. The kinetics of electrode reaction was studied by Tafel slope a, which was obtained by fitting the Tafel function of η=b lgj+a (where η is the electrode potential, b is the Tafel slope; j relates to the current density, and a represents the constant).

### 2.6. Calculation of Electrochemical Active Surface Area (ECSA)

The ECSA was determined using the double layer capacitance method. Generally, the current was measured from CV curves of the non-Faradic potential region obtained at different scan rates shown in Appendix A. The relation between the current difference value $\Delta j=j_{a}-j_{c}$ at a specific potential (1.046 V) and scan rate (v) were fitted linearly. The slope of the fitted lines is the double-layer capacitance ($C_{dl}$). The ECSA can be estimated from double-layer capacitance according to $\mathrm{ECSA}=C_{dl}\cdot A/C_{s}$, where A is equal to the area of the electrode, 1 cm^2^, and C_s_ is the ideal planar capacitance of a smooth surface made of the same materials, taken here as 0.04 mF cm^−2^.

## 3. Results and Discussion

The PBA graphene composite was prepared in two steps: co-precipitation and water bath reduction. The process of preparation is shown in Figure 1a. First, Fe groups (Fe(CN)_6_^3−^) were adsorbed on the negatively charged functional group-rich surface of GO. Then Ni^2+^ ions were quickly added, allowed to interact with Fe(CN)_6_^3−^, and self-assembled into PBA nanoparticles on the GO surface. After the reduction in a water bath at 90 °C, samples were centrifuged to remove the unreacted ions and repeatedly washed with deionized water. After freeze-drying, PBA/rGO composites powder was finally obtained. The morphology of the composite material was obtained by SEM characterization. The co-precipitated PBA material includes irregularly shaped and non-uniform sized particles (Appendix A). During the SEM examination, the surface of material was easily enriched with electric charges, indicating the materials’ poor conductivity. Figure 1b,c show the SEM images of PBA NiFe 2-1/rGO with different magnifications. Comparing the SEM images of the two samples of PBA NiFe 2-1 and PBA NiFe 2-1/rGO, it is obvious that the introduction of graphene to the heterostructure remarkably reduces the size of the surface PBA material and increases the material’s conductivity. PBA samples show obvious diffraction peaks corresponding to the standard PDF card of Prussian blue (JCFDS NO. 01-0239) in XRD patterns (Figure 1e). In addition, the diffraction peaks shifted to a small angle as the proportion of Ni increased, indicating that the excess Ni atoms distorted the PBA lattice and formed the bimetallic NiFe PBA, which is consistent with the literature [17]. In addition to the fainter PBA diffraction peak of the NiFe 2-1/rGO material, a broader diffraction hump peak, which corresponds to the characteristic peak of graphite carbon, appears in the range of 15–27° [30].

To further investigate the structure of the materials, the obtained samples were characterized using Raman spectroscopy. The spectra in Appendix A shows two characteristic Raman vibration modes. The Raman peak centered at 2155 cm^−1^ in the PB sample originates from the Fe^3+^–C≡N–Fe^2+^. However, with Ni incorporation the peak shifts to 2185 cm^−1^ due to the formation of Fe^3+^–C≡N–Ni^2+^ vibrations in PBA samples [31]. Compared with the PBA material, two strong scattering peaks were located at 1345 and 1592 cm^−1^ in the Raman spectrum of the PBA NiFe 2-1/rGO composite sample, which can be ascribed to the D and G peaks of graphene, respectively. The D band mainly represents the defects and disorder of graphene carbon, while the G band corresponds to the sp^2^ carbon vibration [32].

To clarify the chemical bonding of the materials, PBA NiFe 2-1 and PBA NiFe 2-1/rGO were characterized by XPS. The C 1s peak at 284.6 eV was considered a standard for sample charging. The survey spectra of these two samples before OER measurement are shown in Appendix A. To estimate the Ni and Fe species contents, the obtained XPS results were fitted by XPS PEAK software (Figure 2). Two major peaks were found by split-peak fitting located at 855.7 eV for the Ni2p3/2 peak and Ni2p1/2 at 874.0 eV corresponding to the Ni-N coordination in NiN_6_ [19], and the other two peaks (860.62 and 880.09 eV) are the satellite peaks corresponding to the aforementioned two peaks (Figure 2a) [33]. Comparing the Ni 2p high-resolution peak, the Ni 2p 3/2 and 2p 1/2 peak positions are shifted to higher energy for the graphene composite compared with PBA NiFe 2-1, indicating that the interaction between Ni and rGO results in a decreased electron density for Ni and increased the oxidation state of Ni (Figure 2c) [34,35]. The high oxidation state of Ni is favorable for receiving electrons, which contributes to the formation of NiOOH active material, and can increase the charge transfer between the electrode material and the electrolyte [19,36].

Regarding the 2p peak of Fe, two peaks of Fe 2p3/2 and Fe2p1/2 shown in Figure 2b generated by orbital cleavage can be observed at 708.4 and 721.9 eV corresponding to the coordination of Fe^2+^-C [19,37]. Additionally, there is another peak at 711.1 eV, corresponding to Fe^3+^ and indicating the presence of mixed valence states of Fe^2+^ and Fe^3+^ in the composites (Figure 2d) [38]. The appearance of the mixed valence states is mainly attributed to the rGO interface, facilitating the transfer of electrons from Ni^2+^ to Fe^3+^ by the -C≡N- ligand [39]. Meanwhile, the Fe 2p peak in the composite also shifts to higher energy, indicating that the interfacial charge transfer effect promotes the oxidation of Fe, which is consistent with the previous analysis of the Ni 2p peak.

The catalytic performance of graphene itself is weak and almost negligible, and the catalytic activity is primarily derived from PBA materials. By comparison, the catalytic activity of simple PBA NiFe 0-1 is significantly lower than that of PBA NiFe x-1 (x = 1, 2, 3, and 4, respectively) catalysts, and the catalytic activity can be changed by adjusting the Ni/Fe atomic ratio (Figure 3a). However, the poor electrical conductivity of PBA itself will affect charge transfer. Based on the LSV results of PBA NiFe 2-1 and PBA NiFe 2-1/rGO catalysts, it was observed that combining PBA with high electrical conductivity materials can further improve the catalytic activity of PBA materials. Furthermore, the kinetics of the electrode reaction were investigated, based on the equation η=b lg j+a. The Tafel slope b of the material was calculated using LSV results without iR compensation, as shown in Figure 3b. The bimetallic PBA catalyst has faster electrode reaction kinetics than that of the single Fe metal PB material. Additionally, graphene significantly enhances the reaction kinetics of the PBA NiFe electrode material. To more clearly compare the effects of material components and ratios on the catalytic reaction kinetics, the overpotential and Tafel slope values are compared in Figure 3c. The addition of graphene reduces the overpotential of PBA NiFe 2-1 by 25.6 mV (the mass ratio of graphene is not considered here, which means that the effective active catalyst masses of the two electrode materials are different), and the Tafel slope, which measures the electrode kinetics, is reduced by 103.8 mV dec^−1^. Besides the catalytic reaction, durability is a vital indicator of material performance. The PBA NiFe 2-1/rGO material was tested for timing potential at a current density of 30 mA cm^−2^ and it could catalytically precipitate oxygen stably for over 40 h in 1.0 M KOH solution, as shown in Figure 3d. Overpotential, Tafel slope, and durability are vital indicators to evaluate the catalyst performance. Therefore, the parameters of analogous materials in the literature are compared in Appendix A.

The CV curves were acquired at different scan rates in the non-Faraday potential range to clarify the differences in the catalytic performance of the materials (Appendix A). The relationship between current density and scan rate at 1.406 V potential could be well fitted. The material current density and scan rates are both linear, as shown in Figure 4a,b, and the corresponding slope is fitted to obtain the bilayer capacitance value C*_dl_*. Figure 4a depicts the linear fitting results for the PBA materials with varying Ni/Fe ratios, and PBA NiFe 2-1 material has a relatively high C*_dl_* value. The linearly fitted result of PBA NiFe 2-1 and PBA NiFe 2-1/rGO materials are compared in Figure 4b. The presence of graphene significantly increases the C*_dl_* of the material. The electrochemically active area of the electrode material is further calculated, shown in Appendix A, by the equation ECSA = C*_dl_*/C*_s_*, where C*_s_* is the value of planar capacitance, which is taken as 0.04 mF cm^−2^ here [40]. And the electrochemically active area of the graphene composite is substantially increased, since the confinement-growth of PBA nanoparticles on the surface of graphene with significantly reduced size, which enables it to expose more reaction sites in the electrolyte.

To investigate changes in the valence band structure of the materials during the electrocatalytic reaction and clarify the reaction mechanism, the PBA NiFe 2-1 and PBA NiFe 2-1/rGO materials were characterized by XPS after oxygen evolution reaction (OER) (Appendix A and Figure 4c,d, respectively). The two materials still contain the Ni 2p3/2 and 2p1/2 peaks after OER, along with the corresponding satellite peaks. The Ni 2p peak of PBA NiFe 2-1/rGO is negatively shifted after OER process, indicating that the oxygen-containing intermediates transfer electrons to Ni^2+^ during the catalytic reaction, causing Ni^2+^ to be oxidized [38]. Two peaks of Fe^2+^ disappeared in the Fe2p peak, and two peaks corresponding to Fe^3+^ appeared, 710.52 eV and 718.67 eV, indicating that Fe^2+^ was converted to Fe^3+^ during the OER [41,42]. These results suggested that interfacial interactions change the charge density at the metal sites of the material and modulate the reactivity.

## 4. Conclusions

The bimetallic NiFe PBA material self-assembly growing on the surface of rGO by co-precipitation method exhibits excellent electrocatalytic water splitting oxygen evolution properties. Morphology and chemical bonding characterization results of the composite materials show that the introduction of graphene significantly reduces the particle size of the PBA material, and reveal that charge transfer occurs at the heterostructure interface. According to the electrochemical oxygen evolution performance tests, the heterostructure interface effect reduces the oxygen evolution overpotential by 25.6 mV (at the current density of 30 mA cm^−2^), increases the electrode reaction kinetics by 103.8 mV dec^−1^, and increases the electrochemically active surface area of the PBA NiFe 2-1/rGO heterostructure material by 9.6 times compared to PBA NiFe 2-1. This result is primarily attributed to the confinement growth of PBA material on the rGO surface, which increases the electrochemically active area of the material; additionally, the heterostructure interfacial charge transport effect improves the composite’s charge transfer capacity and regulates the activity of the reaction sites.

## Figures and Tables

**Figure 1 molecules-27-05472-f001:**
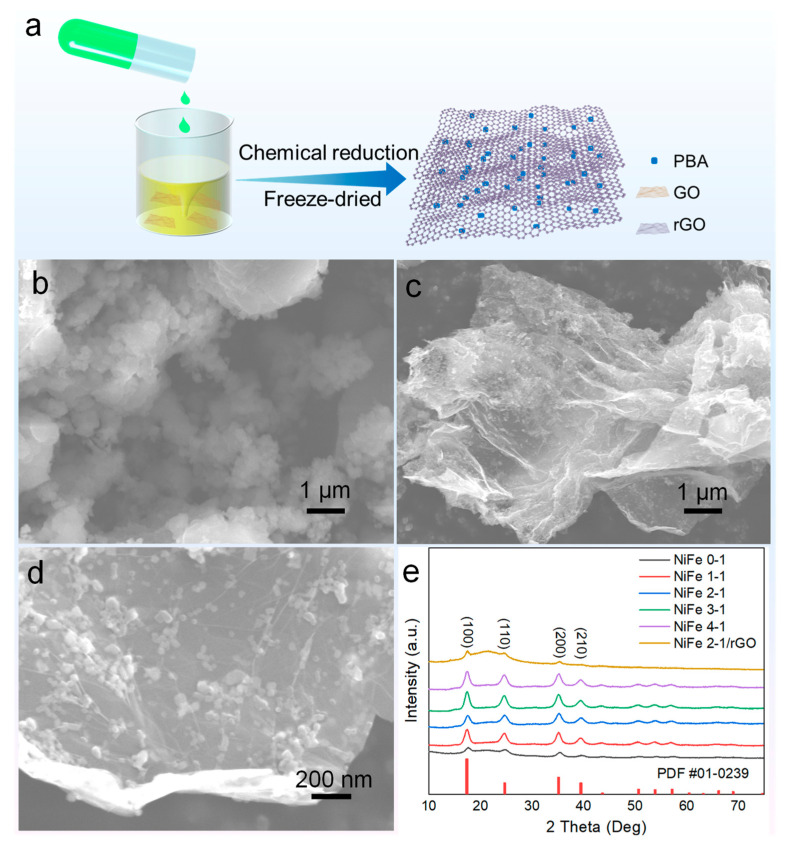
(**a**) Schematic illustration of preparation process; (**b**) SEM images of PBA NiFe 2-1; (**c**,**d**) SEM images or PBA NiFe 2-1/rGO at various magnification; (**e**) XRD patterns of the as obtained PBA samples.

**Figure 2 molecules-27-05472-f002:**
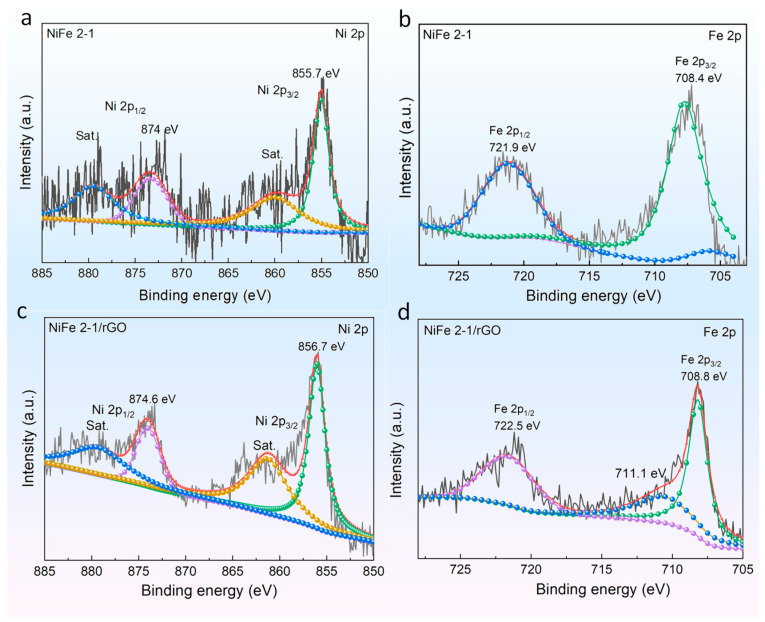
XPS spectra of samples before OER measurement. (**a**,**b**) High-resolution Ni 2p, Fe 2p, and O 1s spectra of PBA NiFe 2-1, respectively; (**c**,**d**) high-resolution Ni 2p, Fe 2p, and O 1s spectra of PBA NiFe 2-1, respectively.

**Figure 3 molecules-27-05472-f003:**
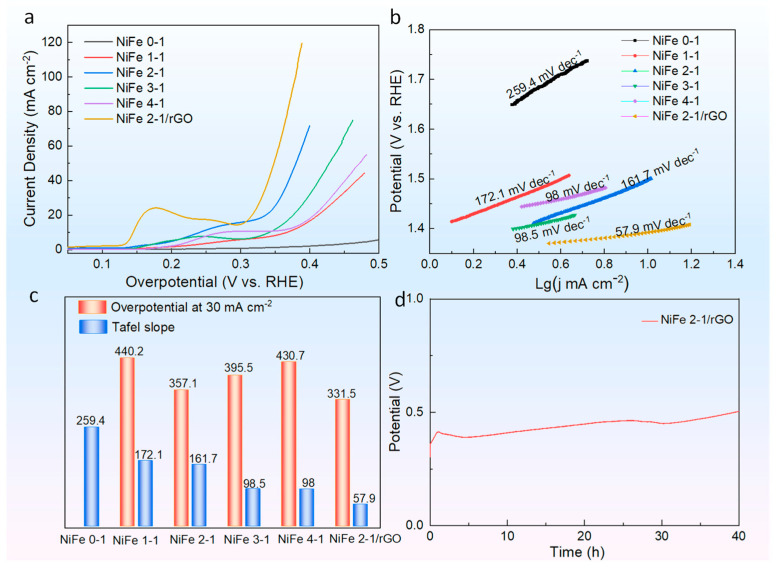
Electrochemical measurement of PBA materials. (**a**) LSV curves; (**b**) linearly fitted Tafel slope; (**c**) histogram of overpotential and Tafel slope values; (**d**) the durability of PBA NiFe 2-1/rGO composite at the current density of 30 mA cm^−2^.

**Figure 4 molecules-27-05472-f004:**
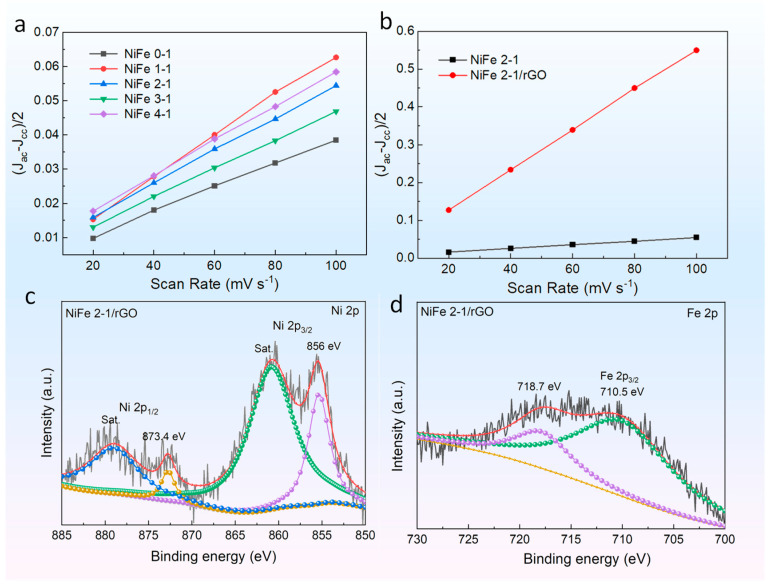
The linearly fitted Cdl results of electrodes. (**a**) PBA NiFe x-1; (**b**) PBA NiFe 2-1 and PBA NiFe 2-1/rGO electrodes; (**c**,**d**) XPS spectra of samples after OER measurement. High-resolution Ni 2p and Fe 2p spectra of PBA NiFe 2-1/rGO, respectively.

## Data Availability

The data presented in this study are available in the article.

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
