# Peer review of "Enhanced Electrochemical Water Oxidation Activity by Structural Engineered Prussian Blue Analogue/rGO Heterostructure"

_molecules, 2022, doi:10.3390/molecules27175472_

Round 1
Reviewer 1 Report
Comments for: molecules-1864268
In this manuscript, the authors reported Prussian blue analogue/rGO Heterostructure catalysts through co-precipitation method. The morphologies, structures and electrochemical catalytic properties were investigated in detail, and ex-situ characterization was also applied to examine the reaction mechanism. I recommend this manuscript to be published after following reversions.
1. Whether GO or rGO is used in the reaction solution? The authors need to clarify in Figure 1e.
2. The authors indicate the presence of graphite carbon by a broader X-ray diffraction hump peak appears in the range of 15-27°. Raman characterization of the material is suggested to further prove the presence of graphene by the characteristic peaks of D and G.
3. Does the GO mentioned in the experimental section stand for graphene oxide? Is the graphene oxide purchased or prepared by authors? The authors should give detail information about graphene oxide.
4. The authors should check and modify some formatting problems in the paper, such as the abbreviation of journal names in the references, according to journal guidelines.
Author Response
Reviewer #1: In this manuscript, the authors reported Prussian blue analogue/rGO Heterostructure catalysts through the co-precipitation method. The morphologies, structures, and electrochemical catalytic properties were investigated in detail, and ex-situ characterization was also applied to examine the reaction mechanism. I recommend this manuscript to be published after the following reversions.
Response: We acknowledge the reviewer for his/her enthusiastic and positive review. We also thank him/her for bringing up new comments, which help us improve our results. In the following, we reply one by one to the referee’s questions.
Comment 1: Whether GO or rGO is used in the reaction solution? The authors need to clarify in Figure 1e.
Response: We are grateful for the reviewer’s time and effort in reviewing our manuscript. The dispersion of graphene oxide (GO) in an aqueous solution is used as the reaction precursor. It is not distinguished in Figure 1e due to our carelessness. With your kind reminder, the figure in the revised manuscript has been updated.
Comment 2: The authors indicate the presence of graphite carbon by a broader X-ray diffraction hump peak appears in the range of 15-27°. Raman characterization of the material is suggested to further prove the presence of graphene by the characteristic peaks of D and G.
Response: We appreciate the reviewer for the professional comment. We took the Raman characterization for the samples in response to the reviewer’s question. The Raman peak centered at 2155 cm-1 in the Prussian blue (PB) sample originates from the Fe3+‒C≡N‒Fe2+. However, the peak shifts to 2185 cm-1 with Ni incorporation due to the formation of Fe3+‒C≡N‒Ni2+ vibrations in PBA samples (Chem. Eng. J. 2020, 397, 125521; Applied Surface Science 2022, 574, 151620). In the Raman spectrum of the PBA NiFe 2-1/rGO sample, two strong scattering peaks locate at 1345 and 1592 cm-1, which can be attributed to the D and G peaks of graphene, respectively (Adv. Mater. 2013, 25, 591-595; ACS Sustain. Chem. Eng. 2019, 7, 13523). The Raman characterization results have been added as Figure S2 to the revised supporting information, and the prime numbers of other figures have been updated accordingly. The analysis of Raman spectra has been added in the revised manuscript on page 10 (marked in yellow).
Comment 3: Does the GO mentioned in the experimental section stand for graphene oxide? Is the graphene oxide purchased or prepared by authors? The authors should give detail information about graphene oxide.
Response: We apologize for our carelessness; GO in the manuscript refers to graphene oxide prepared by the modified Hummer method. The preparation of GO has been supplemented in the revised manuscript’s experimental section on page 5.
Comment 4: The authors should check and modify some formatting problems in the paper, such as the abbreviation of journal names in the references, according to journal guidelines.
Response: We appreciate the reviewer’s detailed comment. We have revised the reference format and other format errors in accordance with the requirements of Molecules. In the revised manuscript, the corrections are highlighted in yellow.
Reviewer 2 Report
Manuscript ID: molecules-1864268
Type of manuscript: Article
Title: Enhanced electrochemical water oxidation activity by structural engineered Prussian blue analogue/rGO heterostructure
Authors: Xiuyun An, Weili Zhu, Chunjuan Tang, Lina Liu, Tianwei Chen, Xiaohu Wang, Jianguo Zhao *, Guanhua Zhang *
In the present work, the authors report the realization of heterostructures based on Prussian blue analogs (PBA) and graphene oxide (rGO) for the electrochemical oxidation of water. The heterostructures were characterized through XRD, SEM, and XPS determination. The electrocatalytic activity was also investigated through cyclic voltammetric analyses. The authors suggest that the confinement-growth of PBA on graphene surface facilitates the electrocatalytic oxygen evolution
The results look reasonable and support the observed trends; the treated arguments are well described and supported by adequate references. However, there are some points not clear that force me to suggest publication on Molecules only after some revisions.
Suggested remarks
Experimental Section
Material Synthesis – The sentence “The resulting mixture was then centrifuged and washed with deionized water to remove unreacted ions. The obtained precipitate was dispersed in 20 ml of deionized water, and 1 ml of 88 mg ml-1 of the ascorbic acid solution was added stirred for 5 min, and then left to react in a water bath at 90°C for two hours.” is somewhat misleading because it is difficult to understand if the product is soluble or not in water….
Calculation of electroactive surface area – Fig. S2 in the SI reports SEM images and not CV curves.
Result and discussion
The authors are invited to explain why they focus their analyses only on the PBA NiFe 2-1 material in terms of the heterostructures realization.
Reviewer 3 Report
Journal Name: Molecules
Title: Enhanced Electrochemical Water Oxidation Activity by Structural Engineered Prussian Blue Analogue/rGO Heterostructure.
The authors prepared the graphene-based heterostructure for energy applications like water oxidation. I recommend its publication with major revision. Here I am mentioning my detailed comments.
1. I suggest authors cite relevant articles on MDPI Molecules so that it will prove the article’s suitability to publish on MDPI molecules.
2. What are the advantages of reduced graphene oxide over doped graphene has to be discussed
3. What is the reason behind selecting 40 hours of OCP has to be discussed
4. Literature review is poor. The surface chemistry of the graphene has to be discussed properly. I recommend authors to read some of the articles mentioned below.
a. Role of defects on regioselectivity of nano pristine graphene
b. Quantum chemical study of Triton X-100 modified graphene surface
c. Regioselectivity in hexagonal boron nitride co-doped graphene
d. Pre-post redox electron transfer regioselectivity at the alanine modified nano graphene electrode interface
https://chemistry-europe.onlinelibrary.wiley.com/doi/abs/10.1002/cctc.201200210 https://www.sciencedirect.com/science/article/abs/pii/S0042207X163019195. The authors also need to discuss the progress in reduced graphene oxide for water oxidation mechanism.
6. The authors need to compare the overpotentials of different graphene materials.
7. The quality of the figures low
8. English has to be improved substantially
9. The error bars should be mentioned and reasons should be discussed for the errors.
10. Graphical abstract is not visible
11. Conclusion has to be improved further
- Please check the grammatical and syntax error
Round 2
Reviewer 3 Report
The authors made sufficient changes I recommend accepting this article in the present form.